# Applications of Noble Metal-Based Nanoparticles in Medicine

**DOI:** 10.3390/ijms19124031

**Published:** 2018-12-13

**Authors:** Bartosz Klębowski, Joanna Depciuch, Magdalena Parlińska-Wojtan, Jarek Baran

**Affiliations:** 1Department of Condensed Matter Physics, Institute of Nuclear Physics Polish Academy of Sciences, 31-342 Krakow, Poland; bartosz.klebowski@gmail.com (B.K.); joannadepciuch@gmail.com (J.D.); 2Department of Clinical Immunology, Institute of Paediatrics, Jagiellonian University, Medical College, 30-663 Krakow, Poland

**Keywords:** nanoparticles (NPs), drug delivery systems (DDS), cancer treatment, radiosensitizer, bioimaging, antibacterial agents

## Abstract

Nanoparticles have unique, size-dependent properties, which means they are widely used in various branches of industry. The ability to control the properties of nanoparticles makes these nanomaterials very interesting for medicine and pharmacology. The application of nanoparticles in medicine is associated with the design of specific nanostructures, which can be used as novel diagnostic and therapeutic modalities. There are a lot of applications of nanoparticles, e.g., as drug delivery systems, radiosensitizers in radiation or proton therapy, in bioimaging, or as bactericides/fungicides. This paper aims to introduce the characteristics of noble metal-based nanoparticles with particular emphasis on their applications in medicine and related sciences.

## 1. Nanoparticles—General Information

Nanoparticles are defined commonly as solid, colloidal particles with sizes ranging from 10 to 1000 nm. However, European and other international committees narrow the above definition to structures having their three dimensions in the order of 100 nm or less. They are offering advantages not possessed by larger particles, for example increased surface to volume ratio or improved magnetic properties [1,2].

Nanotechnology concerns the design, creation and use of materials whose basic unit of measure is a nanometer. This term includes, among others, biomedical and pharmaceutical sciences, physics, advanced materials, and chemistry. The possibility of obtaining sub 100 nm materials, yields their numerous applications in the field of biomedicine, such as imaging agents, drug delivery carriers (chemotherapeutics, genes, proteins etc.) or radiosensitizers in radiation, proton, or photodynamic therapy [3,4,5]. The advantages of noble metal-based nanoparticles, which are important for medical applications, include high biocompatibility, stability and the possibility of large-scale production avoiding organic solvents and thus giving a positive effect on biological systems. These nanoparticles can allow for the controlled release of different drugs. What is more, it is possible to freeze dry nanoparticles to form a powder formulation [6]. For comparison, magnetic nanoparticles, such as iron or cobalt nanoparticles, often require coverage to improve their biocompatibility and prevent against agglomeration, oxidation, or corrosion [7].

There are a lot of clinically approved nanoparticle-based therapeutics, the most common are based on liposomal (Abelcet, AmBisome, DaunoXome, Myocet, Caelyx) or polymeric platforms (Copaxone, Adagen, Macugen, Naulasta, Ranagel) [8]. There is also a list of nanogold platforms-based clinical trials, e.g., AuroLase, NU-0129, or CNM-Au8 [9].

Noble metal-based nanoparticles can be prepared using several methods. One of the most popular methods is chemical reduction of the noble metal precursor (e.g., chloroauric acid, silver nitrate, or chloroplatinic acid to obtain gold, silver, or platinum nanoparticles, respectively) using a proper reducing agent, which can also act as a stabilizer (e.g., sodium citrate in the case of nanogold synthesis) preventing the agglomeration of the nanoparticles [10,11,12]. In addition, by changing reaction parameters, such as temperature, amount of precursor and reducing agent, reaction time, etc., it is possible to synthesize nanoparticles with various shapes and sizes [13].

Recently, green chemistry methods have become more and more popular, allowing for limiting the usage of environmentally harmful substances. For this purpose, reducing agents from green sources are used, e.g., lactic acid, citrus fruits, coffee seeds, etc. [14,15,16]. More sophisticated approaches to obtaining these nanoparticles involve sonochemical, microwave assisted and electrochemical methods [17,18,19].

## 2. Drug Delivery Carriers

### 2.1. Nanoparticles for Drug Delivery

Drug targeting of specific organs and tissues has become a key challenge in recent years. This is due to the fact that in the case of conventional drug administration routes, there are difficulties in reaching the target with the desired dose during a defined period of time. Currently, thanks to the use of modern forms of drugs, so-called drug delivery systems (DDS), it is possible to overcome the problem of biochemical barriers in the body (e.g., brain blood barrier). The novel systems can solve difficulties related to drug solubility and protect the drug from photodegradation and pH changes [20]. The most popular forms of drug delivery include liposomes, liquid crystals, dendrimers, cyclodextrins, micelles, polymerosomes, hydrogels, and nanoparticles (nanospheres and nanocapsules) [21,22,23,24].

Among all types of noble metal-based nanoparticles, gold nanoparticles (Au NPs) have shown a great capacity for use as potential drug delivery carriers (Figure 1). It is relatively easy to obtain Au NPs with various sizes (1–100 nm) and shapes (spherical, rod-like, cage-like, etc.). Moreover, Au NPs can be easy functionalized with different types of molecules. Biocompatibility, stability, macroscopic quantum tunneling effect and presence of surface plasmon resonance (SPR) bands are also important features of Au NPs. Unfortunately, Au NPs are not biodegradable, and their surface modification can change toxicity, biodistribution, and pharmacokinetics of the transported drugs [25,26].

Conjugation of Au NPs with other drugs (i.e., antibiotics or anticancer drugs) is not a problem, but it should be remembered that the biofunctionalization, in order to successfully attach the desired drugs, can change toxicity of nanoparticles. For example, Gu et al. have shown that Au NPs conjugated with polyethylene glycol (PEG) and 3-mercaptopropionic acid do not induce cytotoxicity in cervical cancer cells line (HeLa) [27]. For comparison, Wójcik and co-authors, using the MTT (3-(4,5-dimethylthiazol-2-yl)-2,5-diphenyltetrazolium bromide) test, have confirmed that glutathione-stabilized gold nanoparticles modified with non-covalently bound doxorubicin are more active against feline fibrosarcoma cell lines that unmodified Au NPs [28].

Bhattacharya and co-authors discovered that Au NPs interact with folic acid PEG-amine, which enabled delivery of medicines to tumors via ligation to the folate receptor found on the surface of some cancer cells [29]. Tran and co-authors have obtained, using a one-pot synthesis, 3 nm and 20 nm methotrexate-conjugated gold nanoparticles (MTX-Au NPs). Methotrexate (MTX) is a cytotoxic agent, which is an antagonist of folic acid. Researchers have proven that 3 nm MTX-Au NPs show stronger cytotoxic effect on human choriocarcinoma cell lines compared to free MTX [30].

In other studies, novel doxorubicin-conjugated Au NPs shielded by PEGylation on the surface of Au NPs have been synthesized. This strategy made it possible to obtain a better drug solubility as well as drug release in an acidic environment (Au NPs-Dox-PEG NPs). Using this approach, the intratumoral drug concentration of Au NPs-Dox-PEG NPs was twice as high when compared to free doxorubicin [31]. A similar effect has been achieved in the case of synthesis of Au NPs conjugated with paclitaxel (PTX). This strategy has enabled more efficient anticancer therapy in a murine liver cancer model [32]. Bergen et al. have conjugated Au NPs with galactose-targeting ligand (Figure 2) allowing delivery of a ligand to the asialoglycoprotein receptor. This strategy could be used to treat hepatocellular carcinoma [33]. Recently, Faroog et al. have obtained in one-step synthesis the Au NPs loaded with combination of two anticancer drugs having different mechanisms of action: bleomycin and doxorubicin. The use of such nanohybrids has reduced the systemic drug toxicity and contributed to decrease the possibilities of development of the cancer drug resistance [34]. Silver nanoparticles (Ag NPs) obtained using *Morus alba* leaf extract can be successfully used for the treatment of hepatocellular ailments. The results have showed that Ag NPs exhibit cytotoxic hepatoprotective properties in the rat model [35]. Palladium-based nanoparticles (Pd NPs) may also have potential importance in delivering anticancer drug agents such as cisplatin against A549, SKOV-3, and HepG2 cell lines [36].

### 2.2. Nanoparticles for Gene Delivery

Gene therapy is a method of delivering exogenous DNA or RNA to treat or prevent diseases. Popular viral vectors frequently activate host immune systems reducing the efficiency of gene therapy. The above troubles can be solved by using non-viral systems such as metallic nanoparticles [37].

Recent studies showed that Au NPs with different shapes (e.g., nanospheres or nanorods) protect nucleic acid by preventing DNA or RNA from degradation by nuclease. Au NPs conjugated to oligonucleotides show unique properties that can make them potential gene regulatory agents. These carriers can be divided into covalent (Au NPs may be functionalized with thiolated oligonucleotides) and noncovalent [38]. For example, covalent Au NPs are able to activate immune-related genes in peripheral blood mononuclear cells, but not an immortalized, lineage-restricted cell line. This finding is promising in the application of such conjugates in the development of gene delivery systems [39]. Son and co-authors have attached three fragments of nucleic acids to the surface of Au NPs. In this way, a nanomachine that silenced the polo-like kinase 1 via siRNA was obtained [40]. In turn, Peng and co-authors have synthesized lactoferrin-derived peptides coated Au NPs. The obtained conjugates can efficiently deliver genes encoding vascular endothelial growth factor (VEGF) inducing blood vessel formation [41].

### 2.3. Nanoparticles for Protein Delivery

There is a growing list of evidence documenting the application of nanoparticles as protein carriers. Organothiol, a molecular probe, could be used to study the structure and the morphology of proteins attached to Au NPs [25]. Joshi et al. have obtained insulin-functionalized Au NPs, which have been found useful in transmucosal delivery of drugs for the treatment of diabetes in rat models [42]. Enhancement of insulin delivery efficiency can be achieved by covering Au NPs with a non-toxic biopolymer, such as chitosan, which strongly adsorbs insulin on their surface. Schäffler and co-authors have dealt with conjugation of Au NPs, either with human serum albumin or apolipoprotein E. The results of these experiments showed that the attachment of proteins reduces liver retention compared to traditional citrate-stabilized Au NPs [43]. Rathinaraj et al. obtained herceptin (anti-HER-2/neu monoclonal antibody) immobilized on 29 nm Au NPs improving the interaction of this drug with the suitable receptors on the surface of the breast cancer cells (SK-BR3) [44]. Ag NPs have also been used as protein carriers. For example, Farkhani et al. have combined Ag NPs with cell penetrating peptides (CPP). CPP increase the penetration Ag NPs across the cell membrane causing a reduction of survival breast cancer cells (MCF-7 cell line) [45]. Di Pietro and co-authors have functionalized Ag NPs with a specific peptide sequence consisting of arginine, glycine, and aspartic acid (RGD), allowing for the effective entry of Ag NPs into leukemia and neuroblastoma cells [46].

Numerous applications of noble metal-based nanoparticles as biologically active compounds carriers, give hope for more effective treatment of cancer and other civilization diseases. However, the main difficulty in using these nanoparticles in vivo, are problems with their degradation and elimination from the body. Therefore, the improvement of the pharmacokinetics of such nanoparticles should be the main goal of scientists considering this issue.

## 3. Nanoparticles in Radiation-Based Anticancer Therapy

Radiation therapy (RT) is, next to chemotherapy and surgery, a commonly used method in the treatment of various types of cancer. To improve efficiency of radiotherapy, some small molecules (oxygen and its mimics, gemcitabine, capecitabine), macromolecules (microRNAs, some proteins and peptides), and nanoparticles can be used [47].

The main goal of radiation or proton therapy is the delivery of a destructive dose of radiation to cancer cells with a simultaneous protection of the surrounding healthy tissue. It is possible to achieve this goal by two methods: conforming the dose to the tumor volume or enhancing the sensitivity of the cancer cells to radiation. Gamma and X-rays are characterized by exponential dose deposition with tissue depth, so part of the radiation dose is delivered in front of or behind the tumor, which is a disadvantage of RT [48].

High-energy ionizing radiation such as X-rays is mainly used to ionize cellular organelle or water. Water, the main component of the cell, is the main target of the ionizing radiation. As a result of this interaction lysis of the water molecules occurs. This process, called radiolysis, causes the formation of free radicals, such as the hydroxyl and hydrogen radicals, as well as charged water species. The interaction of free radicals with DNA and cellular structures induces apoptosis. It has been reported that hydroxyl ions are major sources of cellular damage by lipid peroxidation [49].

High atomic number nanoparticles, such as Au NPs, are able to increase the production of secondary electrons or reactive oxygen species (ROS), improving the efficiency of RT [48]. Using this approach, the total dose of radiation can be reduced by nanoparticles, and increases the dose administrated locally to the tumor. Moreover, the side effects can be reduced as well [50].

The photoelectric and Compton effects are the main physical phenomena occurring between X-ray beams and metal-based nanoparticles. A photon is fully or partially absorbed by the nanoparticles, which causes the removal of an electron from the surface of nanoparticles. The ionizing radiation has enough energy to separate at least one electron from the nanoparticle, resulting in ion production. Charged particles are directly ionized because they can interact directly with atomic electrons through Coulomb forces and transfer part of their kinetic energy [51].

In turn, photons are not charged, and hence, are more penetrating. They are not directly ionized, but have enough energy to free an orbital electron, the Compton electron, which is directly ionized. Photoelectric absorptions lead to an increase of the absorbed dose. Auger electrons are produced using energy released from electrons, which fall down from higher orbits during a process of replacement of ejected electrons. Auger electrons can produce a higher density of ionization at a localized area, so they can deposit their energy within the vicinity of the Au NPs. This can lead to a high non-homogenous dose distribution in the nanoscale. It may suggest that the combination of RT and Au NPs, could lead to the enhancement of radiosensitization [52]. The interaction of X-rays with high-Z nanoparticles has been summarized in Figure 3.

The size of nanoparticles used for radiosensitization affects their interactions with biological systems, as well as radiation. Thus, it is necessary to avoid accumulation of Au NPs in organs such as the heart or liver. Otherwise, severe side effects may occur. Akhter et al. have shown that toxicity of Au NPs is minimal, if their sizes very between 5 and 50 nm [53]. The size of nanoparticles is important from the point of view of interaction with radiation. When Au NPs become larger, more of the ionizing events from interaction with secondary electrons and radiation occur in the bulk of the NPs. As a result, the dose deposited in the surroundings of the Au NPs is reduced [54].

Concerning the charge of nanoparticles, there are few papers indicating that positively charged nanoparticles significantly improve cell uptake interacting with negatively charged cell membrane. Moreover, positively charged nanoparticles can selectively target cancer cells because of the presence of highly negatively charged glycocalyx on the surface of some cancer cells [55,56].

Silver nanoparticles also exhibit radiosensitizing properties. Their antitumor mechanism is based on the induction of apoptosis, activating oxidative stress and increasing the fluidity of the cell membrane. The radiosensitization mechanism of Ag NPs is probably related to the release of an Ag^+^ cation from the silver-based structures inside the cells, which penetrated the cells. This cation is able to capture an electron and acts as an oxidizing agent, causing ROS production [57]. In corroboration, Liu and co-authors have shown an increased radiation effect using Ag NPs in the treatment of glioma obtaining anti-proliferative and pro-apoptotic effect [58].

Proton therapy (PT) is another method of radiation-based treatment. In this case the tumor is treated with a focused proton beam, allowing the tumor to be more precisely irradiated, and at the same time, healthy tissues are spared. The main advantage of PT over RT, is that the maximum dose of protons occurs at a certain depth depending on the energy of the beam (so-called Bragg Peak, Figure 4), but not on the patient’s skin. In addition, the dose after reaching its maximum quickly falls to zero, enabling very precise irradiation of the tumor, while saving healthy tissues. Protons have similar biological effects to X-rays used in RT because the relative biological effectiveness (RBE) of the protons is approximately 1.1. RBE is defined as the ratio of a dose of standard radiation to the dose of test radiation necessary to obtain the same biological effect. The reference radiation is usually ^60^Co photons. Additional injection of non-toxic, high-Z nanoparticles into tumor tissue can increase the absorbed dose by up to 6% [59,60].

Sotiropulos and co-authors have shown that the degree of single and double stranded DNA damage in Au NPs-assisted PT depends on the size of nanoparticles, biodistribution, concentration, and energy of protons [61]. They also reported on the efficacy of platinum (Pt NPs), silver, and gadolinium (Gd NPs) nanoparticles as radiosensitizers in hadron or proton therapy. Schlathölter et al. have proven that ~3 nm Pt NPs and Gd NPs nanoparticles are most useful in proton therapy; however, Pt NPs turned out to be more effective [62,63,64].

There are also prerequisites for the usage of nanoparticles in photothermal (PTT) and photo- dynamic therapy (PDT, Figure 5).

PTT uses near-infrared laser photoadsorbers, which generate heat upon near-infrared (NIR) laser irradiation [65]. In turn, PDT is based on the administration of photosensitizing agents, followed by irradiation at a suitable wavelength. Consequently, the released singlet oxygen is toxic to cancer cells [66].

Au NPs are especially desirable in PTT because they can give a strong heating effect upon laser irradiation. This is due to enhanced absorption induced by localized surface plasmon resonance (SPR) [67]. The four most common forms of Au NPs used in PTT are: silica core/gold shell nanoshells (clinical trials), gold nanorods, hollow gold nanocages, and nanostars (all in preclinical state) [68]. Camerin et al. have carried out studies using Au NPs conjugated with phthalocyanine (PC). These conjugates were effectively taken up by amelatonic melanoma cells (B78H1 cell line) and resulted in more than a double increase of cell-death compared to free PC in PDT [69]. Epidermal growth factor peptide-targeted Au NPs are an innovative approach to delivering PC to cancer cells, increasing the efficiency of PTD [70]. In addition to Au NPs, it is also possible to use for PTD Ag NPs or Pt NPs [71,72].

As shown above, noble metal-based nanoparticles have an enormous potential in improving cancer treatment. It would be very interesting to check whether simultaneous treatment of tumor cells with two (or more) types of nanoparticles would give a synergistic effect. The synthesis of noble metal-based nanocomplexes (e.g., consisting of gold and platinum) also seems promising. Perhaps a spectacular result would also be reached via the study of the simultaneous effect of noble metal-based nanoparticles and a radiosensitizing agent of another type (e.g., gemcitabine).

## 4. Bioimaging and Molecular Diagnosis

The development of nanotechnology resulted in improved sensitivity, specificity, and multiplexing of molecular diagnostics. High atomic number nanoparticles are often characterized by intense plasmon resonance driven absorption and scattering properties [25].

X-ray computer tomography (CT) is a commonly used diagnostic tool offering broad availability and quite low costs. The basis for imaging using this method is the fact that healthy and diseased tissues have different density, and as a result, a contrast is generated. By using contrasting agents, an enhancement in the contrast between normal and cancer cells can be obtained [73]. For example, Cheheltani et al. have synthesized small (<5.5 nm) Au NPs, which were encapsulated in poly di(carboxylatophenoxy)-phosphazene nanospheres (Figure 6). The obtained platforms, characterized by biocompatibility and biodegradability, were found to produce intensive CT contrast [74]. Others have used UM-A9 antibodies conjugated with gold nanorods. These gold nanoprobes have shown selectivity towards tumors, giving a better contrast in CT. The above tools can help to detect cancer at the cellular level [75]. Hainfeld et al. used small Au NPs (1.9 nm) as an X-ray contrast agent for the detection of tumors in mice. The results of the experiments showed a lack of Au NPs in the blood after 24 h and simultaneously revealed their significant accumulation in the kidney (10% of the injected dose/g, ID/g), tumor (4% ID/g), liver (3% ID/g), and muscle (1%) after just 15 min. The Au NPs were cleared by the kidneys and did not concentrate in the liver presumably because of the small size [76]. Ag NPs can also be an alternative to traditional contrast agents such as iodine. Indeed, Karunamuni et al. have received promising results using Ag NPs for the dual-energy X-ray breast imaging [77].

Recently, gold nanorods have been conjugated to HER81 monoclonal antibodies binding with high efficiency to Her2 receptors on the surface of a SKBR3 breast cancer cell line. Although promising, it is not known whether these probes will be able to efficiently enter tumor tissue in vivo [78]. Gold nanoparticles conjugated to monoclonal anti-epidermal growth factor receptor (anti-EGFR) monoclonal antibodies bound to tumors with a six times higher affinity than normal cells [79]. Au NPs and Gd NPs were used to detect transplanted human neural stem cells using magnetic resonance (MR), exhibiting improved T1 relaxivity and cell uptake. The resulting conjugates enabled the detection of 70% of the transplanted cells [80]. Blythe and Willets have examined high-resolution fluorescence imaging as a strategy for imaging fluorescently labeled ligands on the surface of Au NPs. Using this approach, it was possible to reconstruct the orientation and shape of the Au NPs substrates and reveal the inhomogeneity in the binding of DNA across the Au NPs surface [81].

Nanoparticles can be also used to locate a target or monitor some intracellular processes. The techniques, such as two-photon-induced photoluminescence (TPL), allow for the detection of Au NPs with various shapes (spherical, rod, and urchin) in hippocampal neurons. An important observation is that the cells may have selectivity towards some nanoparticle geometries [82]. Polat et al. have treated MCF-7 cancer cells with an anti-Her2 monoclonal antibody conjugated to the surface of Au NPs (nanospheres, nanorods, and nanocages) followed by UV-Vis spectroscopy and dark-field microscopy analysis of cellular uptake and bioimaging efficiency [83]. It was found that nanospheres have the most adequate shape for cellular internalization. In turn, nanorods and nanocages are more suitable for imaging. Confocal Raman microscopy supported by gold nanoparticles has also been studied as a promising technique for observing the accumulation of such nanoparticles in different parts of the cell [84].

Oligonucleotide-based probes used to visualize intracellular RNA are highly important tools for measuring biological activity in many systems. In this case, a problem with the penetration of the probe’s molecules into the cell appears, and the exposition of the oligonucleotide-based probes to nucleases, reduces their usefulness. To overcome this limitation, nano-flares consisting of gold nanoparticles functionalized with nucleic acid aptamers with affinity for adenosine triphosphate (ATP) have been synthesized. Thanks to these structures, it became possible to distinguish different types of cells based on the gene expression profile using flow cytometry [85]. Indrasekara and co-authors have obtained a new type of tags to detect cancer cells based on a surface enhanced Raman spectroscopy (SERS) method (Figure 7). In their system, synthesized gold dimers, connected by biphenyl-4,4′-dithiol (DBDT), resulted in a more pronounced signal and higher cytotoxic activity compared to traditional Au NPs. This indicates that SERS-based imaging can be more sensitive than commonly used fluorescence imaging [86]. In another study, Wu et al. have conjugated DNAzyme onto 13 nm Au NPs, obtaining the first sensor of this type to detect metals in living cells [87].

Nanoparticles can also be theranostic agents, which simultaneously enable diagnosis and therapy. For example, Liu and co-authors have used gold nanoparticles for imaging (SERS, CT, and TPL imaging) and photothermal therapy in mouse models of sarcomas and shown that gold nanostars are more effective than gold nanoshells [88]. There are also reports on the theranostic application of Ag NPs or Pt NPs [89,90].

Useful features of noble-metal based nanoparticles, such as ease of surface functionalization, non-cytotoxicity, and optical properties, make them promising agents for diagnostics. On the other hand, it is necessary to overcome certain imperfections, such as the risk of their aggregation, toxicity, or non-specific binding [91]. A big obstacle is also the fact that nanoparticles can be opsonized after getting into the bloodstream. Moreover, the presence of biological barriers could also reduce or even prevent effective diagnostics using nanoparticles. Therefore, the search for new ways to solve these problems is critical.

## 5. Antibacterial, Antiviral, and Antifungal Agents

Among noble metal-based nanoparticles, Ag NPs are the most widespread as antibacterial agents. Already in the nineteenth century, silver salts were known as agents for the treatment of bacterial infections such as ulceration, burns, chronic wounds, or conjunctivitis. Currently, Ag NPs are increasingly used as bactericides due to the acquisition of antibiotic resistance by numerous bacteria [92]. Indeed, antibiotic resistant bacteria (AMR) are a major threat globally and it has been predicted that by 2050, 10 million people will die a year worldwide due to AMR [93]. Among AMR, methicillin-resistant *Staphylococcus aureus* and carbapenemase-producing *Enterbacteriaceae* are particularly dangerous [94,95].

There are several mechanisms explaining the bactericidal effect of Ag NPs. Yamanaka et al. have shown that Ag NPs bind to the cell wall of bacteria or fungi, causing damage to the cell membrane structure, leakage of intracellular components, and finally, cell death [96]. In addition, silver ions can cause oxidative stress by producing free radicals [97]. Another mechanism is based on the destructive binding of Ag NPs to genomic DNA, preventing correct replication [98]. It is also possible to release silver ions by Ag NPs, causing a decrease in the activity of enzymes and other proteins at the transcription level [96]. Shrivastava et al. have developed Ag NPs with an enhanced potency against Gram-negative bacteria. This effect was explained by the contribution of Ag NPs to dephosphorylation of tyrosine residues of putative peptide substrates critical for cell viability and division, resulting in bacteria death [99].

Our group has synthesized Ag NPs using green chemistry with a camomile extract. We have observed a synergistic effect of these two bactericidal factors against four bacterial strains: *S. aureus*, *P. aeruginosa*, *B. subtilis*, and *E. coli* [100]. In another study, we have obtained Ag NPs using clove extract and shown that such Ag NPs have more potent bactericidal and antifungal activity compared to pure clove extract [101]. Similarly, Senthil et al. have synthesised Ag NPs using an ethanol extract of fenugreek leaves. These nanoparticles were shown to be highly toxic against *E. coli* and *S. aureus* [102]. In another study, Holubnycha et al. have developed chitosan-Ag NPs to fight against methicillin-resistant strains of *S. aureus*, providing potentially effective tools for their elimination [103]. Unfortunately, the initial enthusiasm has been dampened by the observations that some Gram-negative bacteria (*E. coli* 013, *P. aeruginosa* CCM 3955, and *E. coli* CCM 3954) can acquire resistance to Ag NPs after repeated exposure. This is a result of bacterial production of the adhesive flagellum protein flagellin, which contributes to the aggregation of Ag NPs [104]. Au NPs are definitely less prevalent in application as bactericidal agents, but there are also promising reports. For example, Shamaila and co-authors have shown that Au NPs obtained using chemical synthesis can be potentially used as antibacterial agents, especially against Gram-negative bacteria. The size of Au NPs also influenced the toxicity, but it varied depending on the type of microbes [105]. Mohamed et al. have synthesized Au NPs and checked their activity against *C. pseudotuberculosis*, a pathogen of goats and sheep. The authors have observed that Au NPs penetrated the cell wall of bacteria and accumulated as agglomerates [106]. Another group has compared antimicrobial activity of Au NPs and Ag NPs dispersed on TS-1 (titanium silicate-1) silicate against *E. coli* and *S. typhi* in water. Ag NPs have turned out to be more effective biocides [107]. Also, Pt NPs possess bactericidal activity, which was shown against *E. coli* and *A. hydrophila* using zebrafish as animal model [108].

Ag NPs also have fungicidal properties; however, they are not as strong as the bactericidal properties described above. Furthermore, the mechanism of Ag NPs interaction with fungi is not well known. For example, *Trichosporon asahii* is a fungal pathogen that became drug resistant recently. It can cause a deadly disease, trichosporosis, so new drugs based, e.g., on silver nanoparticles are highly required [109]. Elgorban et al. have used Ag NPs to remove the plant pathogen *Rhizoctonia solani*. The results of the experiments revealed high fungicidal activity of Ag NPs, even at low concentration [110]. Moreover, Khatami and co-authors have obtained 12 nm Ag NPs using green synthesis with pine pollen and then used them to combat *Neofusicoccum parvum.* These Ag NPs showed a strong effect on the quantitative inhibition of fungi growth [111]. Others have synthesized Ag NPs-chitosan nanocomposites, which were later tested as fungicides against *Candida albicans*. The results showed that these structures are often more active than traditional antifungal drugs [112]. In turn, Wani and co-authors have obtained gold polyhedral nanoparticles as well as gold nanodiscs and checked their antifungal activity against various *Candida* species. The results have revealed, that the gold nanodiscs displayed the stronger anti-fungal activity [113].

There is also a growing list of publications confirming the antiviral properties of noble metal-based nanoparticles, but almost all of them concern Ag NPs. So far, little effort has been made to understand mechanism of the interaction of Ag NPs with viruses. Lu and co-authors have synthesized 10 nm and 50 nm Ag NPs and observed that these nanoparticles inhibit the production of hepatitis B virus-RNA in vitro, and larger nanoparticles showed better activity. The authors suppose that antiviral mechanism of action of Ag NPs is associated with the interactions with the double-stranded DNA of the virus [114]. Speshock et al. have obtained 10 nm Ag NPs and examined them as antiviral agents against Tacaribe virus. They have shown that these Ag NPs are able to inhibit virus activity in the early stages of replications [115]. Another group has revealed the ability of ≈10 nm Ag NPs to inhibit activity of HIV-1 reverse transcriptase preventing multiplication of the virus [116]. On the other hand, 4 nm Ag NPs capped with mercaptoethanol sulfonate can be used to destroy the herpes simplex virus. The authors showed that the mechanism of action of this type of NPs is related to the competition for the binding of the virus to the cell [117].

The increasing resistance of microbes to medicines forces us to develop new effective ways to fight them. Noble metal-based nanoparticles carriers are one of the solutions, which was shown in the examples above. Loading some antibiotics onto such carriers could enhance the bactericidal effect. Another potential solution for improving antibiotic therapy could be, for example, the use of nanoparticles to deliver beta-lactamase inhibitors.

## 6. Summary

The application of nanotechnology has already had a significant impact on medicine and medical sciences. The wide range of applications of noble metal-based nanoparticles, especially Au NPs and Ag NPs, shows that it is worth continuing the work in this area. Huge possibilities of these structures concern their potential use as drug delivery systems, factors improving the quality of radiation-based anticancer therapy, and supporting molecular imaging, as well as compounds with bactericidal, fungicidal, and anti-viral properties. In connection with the above, it is important to implement nanoparticle-based therapies for clinical trials because they can be a perfect tool for diagnostics and the treatment of numerous diseases, with special emphasis on cancer. However, it is necessary to overcome certain barriers resulting from the nature of some nanoparticles, such as problems with biodegradability or porosity. Moreover, there is relatively little information about the toxicity and interaction of these nanoparticles with living normal cells. It is particularly important to improve the pharmacokinetic parameters of the synthesized nanoparticles so they could reach the target site undamaged with high selectivity. This can already be obtained by functionalizing the nanoparticles with the appropriate ligands. Furthermore, a comprehensive approach to sensitizing cancer cells using several types of noble metal-based nanoparticles could yield improved treatment results. There is no doubt that the development of nanotechnology has opened new doors for more effective treatment and diagnosis of various diseases. However, some problems, e.g., the degradation and elimination of metal nanoparticles from the body, remain to be solved.

## Figures and Tables

**Figure 1 ijms-19-04031-f001:**
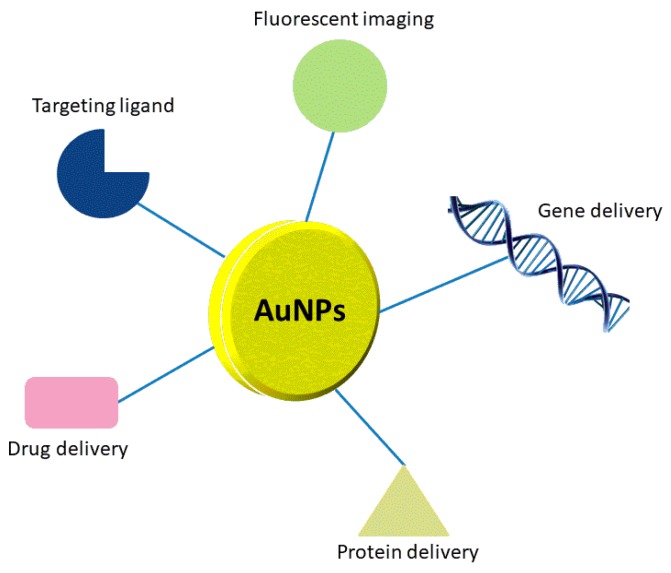
Various applications of Au NPs as carriers.

**Figure 2 ijms-19-04031-f002:**
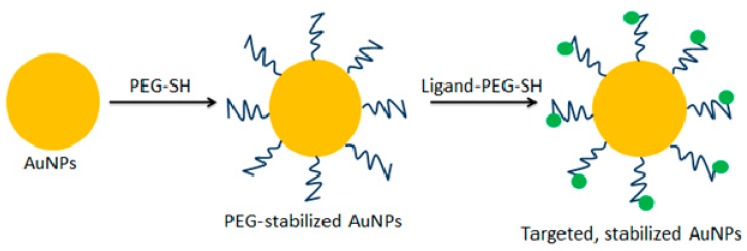
Scheme of the surface functionalization method of Au NPs. Au NPs—yellow circles, PEG-SH—dark blue wavy lines, ligand—green circles (modified from Reference [33]).

**Figure 3 ijms-19-04031-f003:**
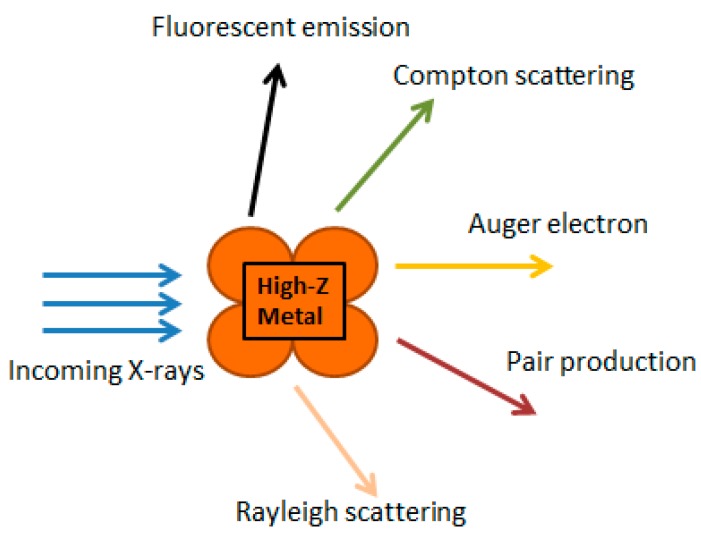
Interaction of X-rays with high-Z material nanoparticles (modified from Reference [49]).

**Figure 4 ijms-19-04031-f004:**
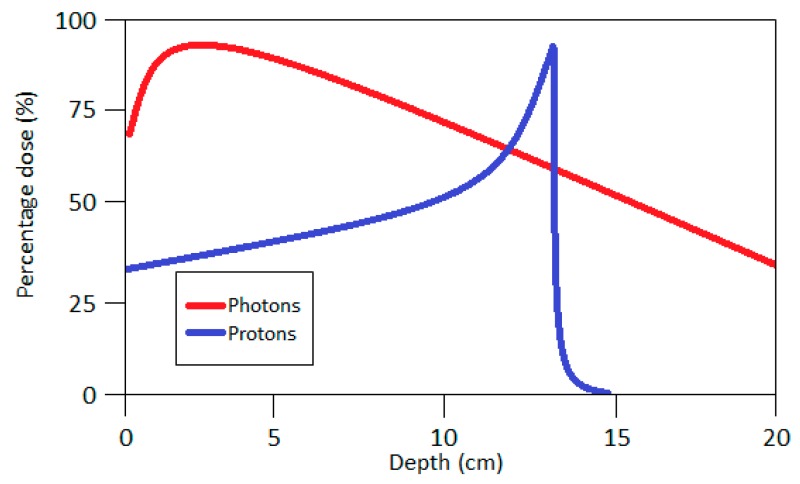
Dose distribution of photons and protons.

**Figure 5 ijms-19-04031-f005:**
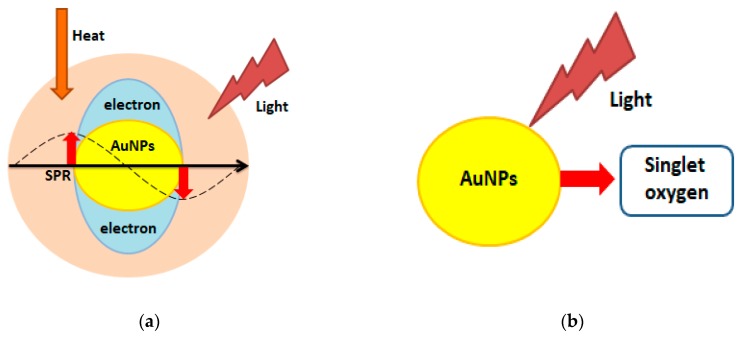
Schematic representation of (**a**) photothermal, and (**b**) photodynamic therapy. (modified from Reference [67]).

**Figure 6 ijms-19-04031-f006:**
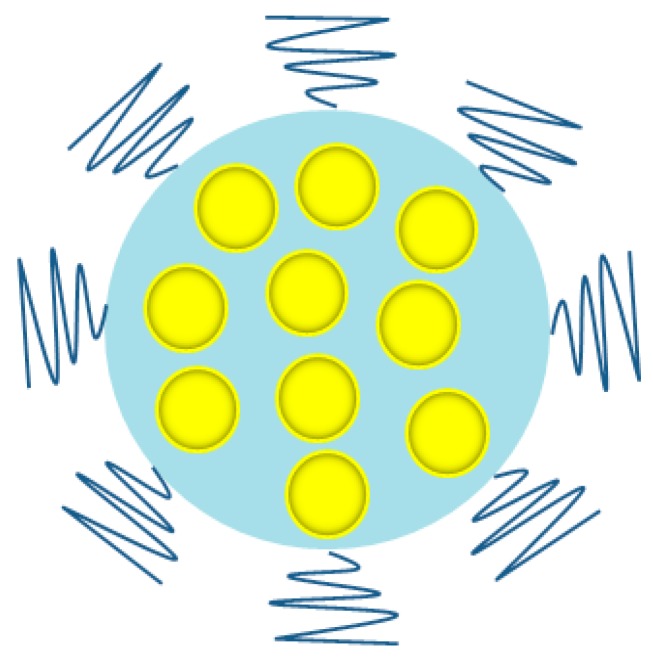
Au NPs encapsulated in polymer nanoparticle. Au NPs—yellow circles, polymer nanoparticle—blue circle, biodegradable polyphosphazene—dark blue wavy lines. (modified from Reference [74]).

**Figure 7 ijms-19-04031-f007:**
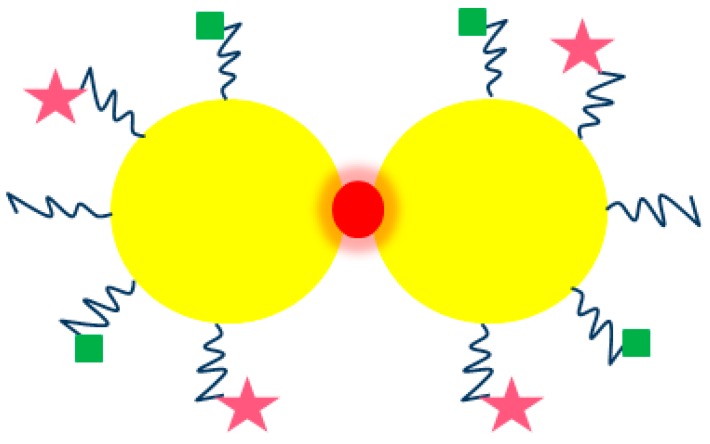
Au NPs dimers. Au NPs—yellow circles, DBDT—red circles, NH_2_-PEG-SH—blue wavy lines, peptide—green squares, and rhodamine—pink stars. (modified from Reference [86]).

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
