# Peer review of "Applications of Noble Metal-Based Nanoparticles in Medicine"

_ijms, 2018, doi:10.3390/ijms19124031_

Reviewer 1 Report
In this mini-review titled, “Applications of noble metal-based nanoparticles in medicine”, Klębowski and co-authors argue the characteristics of noble metal-based nanoparticles and their applications in medicine and related sciences. Therefore they analyze the studies present in literature in the last decade, as can be seen from the extensive biography used in the field of nanotechnology. The described work is interesting and also focuses on different aspects of noble metal-based nanoparticles such as the biodegradation, the surface chemistry, and toxicity. Their application as drug delivery for gene and protein delivery, in radiation-based anticancer therapy and bioimaging have also been considered. There are several issues that need to be addressed:
1. Drug delivery: The authors point out that there are many nanoparticle systems already used in the clinic but they do not show that until now there is no nanoparticles carrier based on noble metals.
2. Drug delivery: Are studies already known on the use of nanoparticles based on noble metal as theranostic agents?
3. Targeted drug delivery: he authors highlight that the conjugation of nanoparticles for the targeted drug delivery also requires a chemical modification of the surface that could change the chemical-physical characteristics of the system and its toxicity.
4. Bioimaging and molecular diagnosis: in this chapter the authors should briefly discuss also the potential advantages / disadvantages of the use of nanoparticles with respect to the diagnostic techniques available today. In fact, AuNPs, of various sizes, and shapes have been rediscovered as new alternatives for fluorescent bioimaging.
Author Response
Reviewer 1.
Drug delivery: The authors point out that there are many nanoparticle systems already used in the clinic but they do not show that until now there is no nanoparticles carrier based on noble metals.
According to the Reviewer’s suggestion we have introduced a required info regarding clinical trials based on noble metals. An appropriate reference has also been included in the main text (line 44-45).
Drug delivery: Are studies already known on the use of nanoparticles based on noble metal as theranostic agents?
Indeed, nobel metals can be used as theranostic agents and we appreciate the Reviewer’s comment. The respective para has been introduced into the main text, according to the Reviewer’s suggestion. However, the required comment was implemented into section 4 of our ms. (Bioimaging and molecular diagnosis; line 320-324) as we think it better fits to the context of this part.
Targeted drug delivery: the authors highlight that the conjugation of nanoparticles for the targeted drug delivery also requires a chemical modification of the surface that could change the chemical-physical characteristics of the system and its toxicity.
Yes, the system has to be biofunctionalized in order to successfully attach the desired drugs and this procedure may change toxicity of nanoparticles, as was evidenced by others. Appropriate references have been included in the main text (line 79-86).
Bioimaging and molecular diagnosis: in this chapter the authors should briefly discuss also the potential advantages / disadvantages of the use of nanoparticles with respect to the diagnostic techniques available today. In fact, AuNPs, of various sizes, and shapes have been rediscovered as new alternatives for fluorescent bioimaging.
The relevant para discussing advantages/disadvantages of the use of nobel metal particles in diagnostics has been introduced into the main text, according to the Reviewer’s suggestion (line 325-331).
We hope, that our answer is complete and satisfactory.
Reviewer 2 Report
This paper aims to introduce the characteristics of noble metal-based nanoparticles with particular emphasis on their applications in medicine as drug delivery systems, radiosensitizers in radiation or proton therapy, in bioimaging or as bactericides/ fungicides.
I found the article suitable for publication. However some aspects must be included in the present text and complete this review. These aspects are detailed below:
In the first section (Nanoparticles-general information), the definition of nanoparticles is a bit confusing. Although nanoparticles are strictly considered particles with sizes between 1-100nm, it is currently acceptable to include submicron systems (<1000 nm). The authors must include an explanation and include bibliographic references.
In this section also it is important to highlight the advantages of the noble metal-based nanoparticles and list most of the advantages and unique properties of the typical noble-metal nanoparticles used in biomedical applications (i.e. Au, Ag, Pt). It would be interesting indicate comparative advantages with other metal-based nanoparticles (i.e. Iron-based nanoparticles).
The authors should include a section (or a paragraph) to describe the most common synthetic methods for obtaining the different nanoparticles (Au, Ag, Pt).
In an exercise of self-criticism, the authors should include at the end of each section (drug delivery, anticancer therapy, bioimaging…) the main drawbacks of the current designs and propose some alternatives to tentatively overcome these problems. Additionally, in the summary section, a list of suggested future improvements for these nanosystems could be added.
In general, I found the review suitable for publication, but some aspects detailed in the present review report should be addressed to implement the manuscript.
Author Response
Reviewer 2
In the first section (Nanoparticles-general information), the definition of nanoparticles is a bit confusing. Although nanoparticles are strictly considered particles with sizes between 1-100nm, it is currently acceptable to include submicron systems (<1000 nm). The authors must include an explanation and include bibliographic references.
We are grateful the Reviewer for pointing-out this discrepancy. We have clarified this issue by introducing the relevant info and reference, according to the Reviewer’s suggestion (line 25-26).
In this section also it is important to highlight the advantages of the noble metal-based nanoparticles and list most of the advantages and unique properties of the typical noble-metal nanoparticles used in biomedical applications (i.e. Au, Ag, Pt). It would be interesting indicate comparative advantages with other metal-based nanoparticles (i.e. Iron-based nanoparticles).
We have improved the entire section according to the Reviewer’s comment. The required para has been introduced into the main text (line 34-41).
The authors should include a section (or a paragraph) to describe the most common synthetic methods for obtaining the different nanoparticles (Au, Ag, Pt).
We appreciate this important comment given by the Reviewer. We have introduced the relevant para into the main text (line 46-57).
In an exercise of self-criticism, the authors should include at the end of each section (drug delivery, anticancer therapy, bioimaging…) the main drawbacks of the current designs and propose some alternatives to tentatively overcome these problems. Additionally, in the summary section, a list of suggested future improvements for these nanosystems could be added.
Indeed, metal nanoparticle based medical applications have several drawbacks and limitations. We have tried to address this issue at the end of each section and in the summary, and to highlight the required improvements of these systems, according to the Reviewer’s suggestion (line 148-153; 249-255; 325-331; 400-404; 418-425).
We hope, that the Reviewer will find our answers complete and satisfactory.